# Consumo Energético de Algoritmos Evolutivos en C++: Impacto del Compilador y el Nivel de Optimización

**Carlos Cotta**[*]
ITIS Software, Universidad de Málaga, España
ccottap@lcc.uma.es

**Jesús Martínez-Cruz**[**]
Dept. Lenguajes y Ciencias de la Computación, ETSI Informática,
Campus de Teatinos, Universidad de Málaga, 29071 Málaga, España
jmcruz@uma.es

## Abstract

Este trabajo estudia el consumo energético de los algoritmos evolutivos (AEs) atendiendo a las decisiones tomadas durante la construcción del programa ejecutable, particularmente la elección del compilador y las opciones de optimización empleadas. El estudio abarca el caso de AEs implementados en C++ y compilados para Windows con dos herramientas distintas: Clang y Microsoft Visual C++. Para cada compilador se consideran cuatro niveles diferentes de optimización en el código ejecutable obtenido. A partir de un banco de pruebas compuesto por funciones de optimización numérica de alta dimensionalidad, se observa cómo hay diferencias muy significativas a favor de Clang tanto en tiempo de cómputo como en energía consumida. Se aprecia también cómo, en términos de energía total, no hay apenas diferencias significativas entre los niveles de optimización más altos para cada compilador, pero sí las hay en el perfil de potencia desarrollada por el procesador y en la temperatura del mismo, atendiendo al compilador empleado.

## 1. Introducción

En los últimos años, el empleo de métodos de inteligencia artificial (IA) está viviendo una fase expansiva sin precedentes que la hace formar parte no ya del ámbito científico sino incluso de la vida cotidiana. Indudablemente, esta explosión lleva consigo un significativo coste ambiental aparejado. Así, de acuerdo con estimaciones de 2023, los centros de datos que ejecutan software de IA masivo producen hasta el 5-9 % de la demanda mundial de electricidad y el 2 % de todas las emisiones de $CO_2$ [11]. Ante estos números, no es de extrañar que sea prioritario el estudio y desarrollo de algoritmos eficientes energéticamente en este contexto.

Si bien esta perspectiva del análisis del rendimiento de los sistemas informáticos está enraizada en las tecnologías de la información desde hace décadas a través de la noción de *Green Computing* [3], las especificidades de la IA la hacen merecedora de una consideración particularizada. Habitualmente, el enfoque seguido durante la investigación y el uso de estas técnicas ha estado orientado a maximizar el rendimiento, ya sea obteniendo mejoras en clasificación, predicción o en optimización de las soluciones, dependiendo del contexto y la técnica particular empleada. Estas mejoras pueden conseguirse en muchos casos a expensas de aumentar exorbitantemente el coste computacional de los métodos empleados [10]. Esto ha llevado a la denominación de *IA roja* a estos enfoques en los

---

[*]ORCID: 0000-0001-8478-7549
[**]ORCID: 0000-0002-8847-8900

XVI XVI Congreso Español de Metaheurísticas, Algoritmos Evolutivos y Bioinspirados (maeb 2025).

que el coste computacional no escala a la par con el beneficio obtenido. Por contra, se denomina *IA verde* a los enfoques en los que los avances se consiguen sin aumentar o incluso reduciendo las externalidades asociadas [14]. El objetivo último es moverse hacia un paradigma de computación en el que se asegure la sostenibilidad en el desarrollo y uso de la IA [17, 18].

El enfoque mencionado tiene una trayectoria que puede considerarse relativamente asentada en el ámbito del aprendizaje automático (véase, por ejemplo, [10, 15]), en parte debido a las características de este paradigma. En el contexto de la computación evolutiva, la situación es mucho más incipiente, si bien se han dado diferentes pasos en esa dirección, fundamentalmente dirigidos al estudio del impacto de la parametrización [1, 6, 8], del entorno de ejecución [13], y de la metodología experimental [4]. En este trabajo, abordamos una nueva dimensión, analizando el impacto que las decisiones tomadas durante la construcción del programa ejecutable (en términos del compilador y las opciones empleadas) tienen en el perfil de consumo energético de un algoritmo evolutivo.

## 2. Materiales y Métodos

Tal como se ha anticipado, el estudio se centra en el papel que el compilador y las opciones de compilación juegan en el consumo energético de un AE. Nuestra hipótesis de partida es que el compilador, como traductor del código fuente a código máquina, es un elemento importante en el ciclo de vida energético del software, y el modo en el que esta traducción se realice puede tener un impacto profundo en el perfil de consumo del programa. Esto es particularmente cierto habida cuenta de que el desarrollador puede además optar por diferentes opciones o niveles de optimización en el código máquina producido, tendientes a priorizar diferentes aspectos como la velocidad o el tamaño. El cómo estas decisiones afectan al consumo energético resulta, por lo tanto, del mayor interés. En esta sección, describiremos los entornos y opciones de compilación consideradas (Sección 2.1), y detallaremos con precisión la metodología experimental empleada (Sección 2.2).

### 2.1. Entornos de Compilación

El estudio se ha realizado sobre una biblioteca de algoritmos evolutivos en C++[3], implementación gemela de una biblioteca análoga en Java, y diseñada con el propósito de ser generalizable y extensible mediante el empleo de patrones de diseño útiles [9] que posibilitan la creación en tiempo de ejecución de los operadores y parámetros específicos del AE a partir de un archivo de configuración JSON.

Para la construcción del ejecutable se ha utilizado una versión reciente de Visual Studio Community 2022 (versión 17.12.4) para Windows, que incorpora tanto el compilador nativo Microsoft Visual C++[4] (*cl.exe*, versión 19.42.34436), como Clang (*clang-cl.exe*, versión 18.1.8). Para ambos compiladores, el entorno de Visual Studio proporciona opciones de optimización comunes:

- Optimizaciones deshabilitadas: `/Od`
- Optimización de tamaño de código: `/O1`. Con esta opción se prioriza la creación de un ejecutable más pequeño frente a la velocidad de ejecución.
- Optimización de velocidad: `/O2`. Esta opción se enfoca a mejorar el rendimiento de la aplicación, aunque es interesante destacar que, para que el compilador de Clang y el de Microsoft realicen optimizaciones comparables, a este último hay que añadirle la opción `/Ot`. Esta opción (`/Ot`) fuerza a generar código más rápido incluso si resulta un ejecutable ligeramente mayor al que se obtendría con la opción `/O2`.
- Optimización de velocidad máxima: `/Ox`. Aquí se habilitan la mayoría de optimizaciones de velocidad comunes a ambos compiladores (excluyendo, por ejemplo, la paralelización de bucles o las aproximaciones en operaciones con flotantes, que podrían impactar en el comportamiento de los algoritmos evolutivos).

La experimentación se ha orientado a analizar el impacto que la selección de estas opciones de compilación tiene en la huella de procesamiento del algoritmo evolutivo. A tal efecto, emplearemos la metodología experimental descrita a continuación.

---

[3]Esta biblioteca está disponible en nuestro repositorio GitHub `https://github.com/Bio4Res/ea-cpp`
[4]De aquí en adelante, nos referiremos a él simplemente como Visual C++

Tabla 1: Formulación matemática de las funciones consideradas en la experimentación

| función | fórmulación matemática |
|---------|------------------------|
| **ackley** | $f(\mathbf{x}) = -a \cdot \exp\left(-b\sqrt{\frac{1}{d}\sum_{i=1}^{d} x_i^2}\right) - \exp\left(\frac{1}{d}\sum_{i=1}^{d}\cos(cx_i)\right) + a + \exp(1)$ 
 $a = 20, b = 0.2, c = 2\pi$ |
| **rastrigin** | $f(\mathbf{x}) = a \cdot d + \sum_{i=1}^{d}\left(x_i^2 - 10\cos(2\pi x_i)\right)$ 
 $a = 10$ |
| **rosenbrock** | $f(\mathbf{x}) = \sum_{i=1}^{d-1}\left[b(x_{i+1} - x_i^2)^2 + (a - x_i)^2\right]$ 
 $a = 100$ |

Tabla 2: Tamaño del fichero ejecutable (en bytes) en función del compilador y del nivel de optimización empleado

| compilador | /Od | /O1 | /O2 | /Ox |
|------------|-----|-----|-----|-----|
| **Clang** | 760320 | 471552 | 553472 | 553472 |
| **Visual C++** | 754688 | 430080 | 470528 | 471552 |

## 2.2. Metodología Experimental

Los experimentos se han realizado considerando un banco de pruebas compuesto por tres funciones de optimización numérica: ackley, rastrigin y rosenbrock – véase la Tabla 1. En todos los casos se ha empleado una versión 100-dimensional de estas funciones. En cuanto al AE, se ha utilizado un modelo generacional elitista estándar (tamaño de población = 100, selección por torneo binario, cruce con el operador BLX-$\alpha$ [7] y mutación gaussiana) con $maxevals = 10^7$ llamadas a la función objetivo por cada ejecución [5].

Se ha construido un programa ejecutable para arquitecturas x64 con los compiladores Clang y Visual C++, empleando tres niveles de optimización de código (/O1, /O2, /Ox) además de una versión con la optimización desactivada (/Od). La Tabla 2 muestra como referencia el tamaño del programa ejecutable final dependiendo del compilador empleado y del nivel de optimización considerado. Para cada una de estas ocho versiones del ejecutable[6] y cada una de las tres funciones del banco de pruebas se han realizado $n = 30$ ejecuciones del AE, dejando 100 segundos de descanso del procesador entre las mismas con objeto de permitir a este volver a un estado de reposo y reducir los efectos histeréticos [5].

Los experimentos se han realizado en un ordenador de sobremesa dotado de un procesador Intel Core i7-9700F con 16 GB de RAM, NVIDIA GeForce GTX 1050 Ti y disco duro Seagate ST1000DM010 2EP102 (SATA III, 6 Gb/s), con Windows 10 Pro (22H2). Las mediciones de energía se han llevado a cabo mediante la herramienta Intel® Power Gadget. Con el objeto de identificar cuál es la contribución real de la ejecución de cada algoritmo al consumo energético de la máquina, determinamos en primer lugar el consumo basal del sistema (medido cuando el ordenador se ejecuta sin ninguna aplicación de usuario al margen del sistema operativo) y lo restamos de las mediciones reales cuando se ejecuta el AE, obteniendo así el exceso de consumo debido a la ejecución del algoritmo.

## 3. Resultados Experimentales

Todos los datos obtenidos en la experimentación están disponibles en nuestro repositorio de datos [7]. Las Tablas 3-4 muestran un sumario de las mediciones de energía y tiempo. Una primera inspección de las mismas arroja dos observaciones importantes: en primer lugar, se aprecia una considerable

---

[5]Los ficheros de configuración empleados están disponibles en nuestro repositorio GitHub `https://github.com/Bio4Res/ea-test-energy`

[6]En realidad, y tal y como se comentaba en la Sección 2.1, junto a /O2 se está utilizando /Ot, aunque Clang no la tiene en cuenta. Por motivos de legibilidad, emplearemos /O2 a lo largo del presente trabajo.

[7] `https://osf.io/rh8pb/`

Tabla 3: Consumo energético (J) de cada ejecución para cada nivel de optimización dependiendo de la función objetivo y el compilador empleado. Cada entrada de la tabla contiene la media y el error estándar de la media en las 30 ejecuciones del algoritmo.

| nivel | Clang | | | Visual C++ | | |
|---|---|---|---|---|---|---|
| | **ackley** | **rastrigin** | **rosenbrock** | **ackley** | **rastrigin** | **rosenbrock** |
| /0d | $4601.9 \pm 2.5$ | $4809.2 \pm 2.2$ | $4950.0 \pm 4.6$ | $4960.2 \pm 2.8$ | $5122.2 \pm 3.4$ | $5227.3 \pm 2.4$ |
| /01 | $589.4 \pm 1.5$ | $631.4 \pm 1.3$ | $593.3 \pm 1.0$ | $1262.4 \pm 3.9$ | $1278.1 \pm 0.7$ | $1237.0 \pm 0.6$ |
| /02 | $550.6 \pm 1.1$ | $617.6 \pm 0.8$ | $544.4 \pm 1.1$ | $1235.9 \pm 0.8$ | $1251.5 \pm 0.8$ | $1217.5 \pm 0.8$ |
| /0x | $548.5 \pm 1.3$ | $616.3 \pm 1.0$ | $545.9 \pm 1.6$ | $1236.7 \pm 1.4$ | $1249.3 \pm 0.7$ | $1220.4 \pm 0.7$ |

Tabla 4: Duración (s) de cada ejecución para cada nivel de optimización dependiendo de la función objetivo y el compilador empleado. Cada entrada de la tabla contiene la media y el error estándar de la media en las 30 ejecuciones del algoritmo.

| nivel | Clang | | | Visual C++ | | |
|---|---|---|---|---|---|---|
| | **ackley** | **rastrigin** | **rosenbrock** | **ackley** | **rastrigin** | **rosenbrock** |
| /0d | $213.1 \pm 0.1$ | $224.4 \pm 0.0$ | $230.3 \pm 0.1$ | $240.7 \pm 0.0$ | $250.6 \pm 0.0$ | $257.4 \pm 0.0$ |
| /01 | $23.6 \pm 0.1$ | $28.2 \pm 0.1$ | $24.4 \pm 0.1$ | $57.8 \pm 0.0$ | $58.4 \pm 0.0$ | $57.2 \pm 0.0$ |
| /02 | $22.6 \pm 0.1$ | $26.2 \pm 0.1$ | $22.0 \pm 0.1$ | $57.1 \pm 0.0$ | $57.6 \pm 0.0$ | $56.6 \pm 0.0$ |
| /0x | $22.6 \pm 0.1$ | $26.1 \pm 0.0$ | $22.0 \pm 0.0$ | $57.2 \pm 0.0$ | $57.7 \pm 0.0$ | $56.6 \pm 0.0$ |

Tabla 5: Comparación vis a vis de los diferentes niveles de optimización en relación a la energía consumida y al tiempo de cómputo. Cada entrada de la tabla contiene 6 símbolos, los tres primeros correspondientes a Clang y los tres siguientes a Visual C++. Dentro de cada grupo, los símbolos corresponden a las funciones de ackley, rastrigin y rosenbrock, respectivamente. El símbolo +/=/- indica que el nivel de optimización de la fila es superior/indistinguible/inferior al de la columna de acuerdo con un test de Wilcoxon ($\alpha = 0.05$).

| | energía | | | | tiempo | | | |
|---|---|---|---|---|---|---|---|---|
| | /0d | /01 | /02 | /0x | /0d | /01 | /02 | /0x |
| /0d | ● | ------ | ------ | ------ | ● | ------ | ------ | ------ |
| /01 | ++++++ | ● | ------ | ------ | ++++++ | ● | ------ | ------ |
| /02 | ++++++ | ++++++ | ● | -===-+ | ++++++ | ++++++ | ● | ====== |
| /0x | ++++++ | ++++++ | +===+- | ● | ++++++ | ++++++ | ====== | ● |

superioridad de Clang sobre Visual C++ en ambas métricas. Si bien en el caso del empleo de la opción /0d ambos compiladores se mueven en el mismo rango aproximado de valores (aunque las diferencias siguen siendo estadísticamente significativas a nivel $\alpha = 0.05$ de acuerdo con un test de rangos de Wilcoxon), en el momento en el que se optimiza el código, Visual C++ es alrededor del doble de ineficiente en energía o tiempo. Esto puede ser debido a diferentes factores dependientes del compilador, tales como el uso de diferentes métodos y heurísticas para optimizar el código, o incluso la mejor adecuación del código fuente original o del procesador destino a dichos mecanismos de optimización.

En segundo lugar, hay una drástica reducción tanto del tiempo de cómputo necesario como de la energía consumida cuando se emplean opciones de compilación para optimizar el código. Esta reducción se observa de manera consistente tanto para Clang como para Visual C++, y tal como se muestra en la Tabla 5 es siempre estadísticamente significativa a nivel $\alpha = 0.05$ de acuerdo con un test de rangos de Wilcoxon. Si bien la reducción en tiempo de cómputo puede ser generalmente esperada, es interesante ver que se traduce también en un menor consumo total por cada ejecución. Lógicamente, este debería ser el caso si la potencia desarrollada por el procesador permaneciera constante, pero como claramente se observa en la Figura 1 no es este el caso. De hecho, puede apreciarse cómo al emplearse optimizaciones de código el procesador trabaja en un régimen de potencia superior (o alternativamente, el código sin optimizar se ejecuta durante más tiempo pero a menor intensidad). Sin embargo, considerando únicamente el consumo acumulado, este perfil más

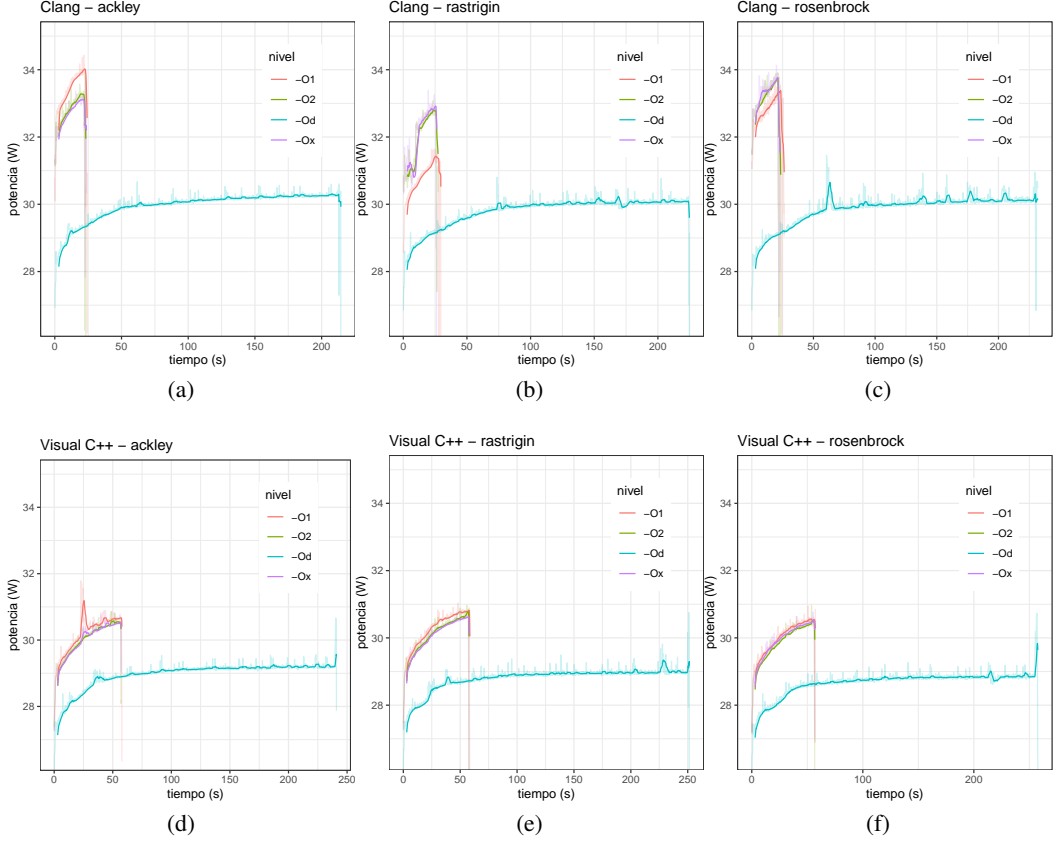

Figura 1: Potencia desarrollada durante la ejecución del algoritmo. La fila superior corresponde a Clang y la inferior a Visual C++. De izquierda a derecha en cada fila: ackley, rastrigin y rosenbrock.

agresivo de uso del procesador resulta ser también más energéticamente eficiente. Puede inferirse que el código sin optimizar conlleva un coste energético de ejecución que lastra el rendimiento global del código con opción /Od.

Un análisis comparativo de las restantes opciones de optimización indica que /O1, si bien notablemente superior a /Od, es ligera pero consistentemente inferior al resto de opciones de optimización (nuevamente, las diferencias son siempre estadísticamente significativas como se ve en la Tabla 5). En este caso, la reducción de tamaño consigue simplificar el código lo suficiente como para que haya una reducción drástica con relación a /Od, pero no alcanza el nivel que pueden proporcionar los niveles de optimización más altos. Precisamente en relación a dichos niveles superiores de optimización, la situación es mucho más matizable. Las diferencias son pequeñas y aunque pueden llegar a ser estadísticamente significativas en algún caso concreto (i.e., para alguna función y compilador específico), no se aprecian tendencias sistemáticas en términos del tiempo o energía acumulada por ejecución. Sensu contrario, los resultados indican que emplear una optimización más agresiva no está suponiendo un coste adicional durante la ejecución del AE. Nótese tanto en relación con este aspecto como con todos los anteriormente mencionados, que la dispersión de los resultados experimentales (véase el error estándar de la media en las Tablas 3-4) es pequeña en relación con las magnitudes observadas, lo que apunta a que el número de réplicas consideradas puede ser suficiente para capturar el comportamiento promedio de los algoritmos.

Relacionado con el empleo de las opciones de compilación más agresivas, un factor a menudo infraconsiderado en el ciclo de vida energético de los algoritmos evolutivos es el propio coste de construcción del ejecutable. Siguiendo la misma metodología descrita en la Sección 2.2, la Tabla 6 muestra el coste en energía y tiempo necesarios para compilar el código de la biblioteca de AEs empleada. Estas mediciones arrojan datos interesantes. En primer lugar, Clang emplea sistemáticamente más tiempo en cada compilación, pero no necesariamente más energía (todas las

Tabla 6: Consumo energético (J) y tiempo requerido (s) en cada compilación del AE, según el compilador y las opciones de optimización. Se muestra la media y el error estándar de la media sobre 30 compilaciones.

| compilador | energía (J) | | | | tiempo (s) | | | |
|---|---|---|---|---|---|---|---|---|
| | /Od | /O1 | /O2 | /Ox | /Od | /O1 | /O2 | /Ox |
| Clang | $118.4 \pm 1.9$ | $193.9 \pm 0.4$ | $224.4 \pm 0.2$ | $224.6 \pm 0.4$ | $5.9 \pm 0.1$ | $9.5 \pm 0.0$ | $11.0 \pm 0.0$ | $11.0 \pm 0.0$ |
| Visual C++ | $121.4 \pm 3.2$ | $218.2 \pm 0.4$ | $221.8 \pm 0.4$ | $221.3 \pm 0.4$ | $5.5 \pm 0.0$ | $7.0 \pm 0.0$ | $7.2 \pm 0.0$ | $7.2 \pm 0.0$ |

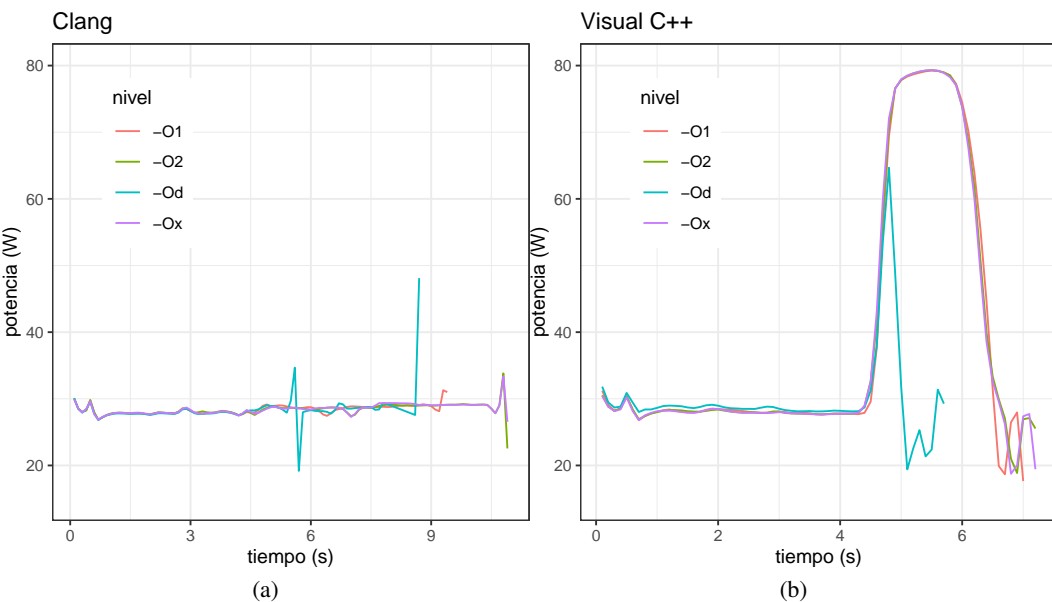

Figura 2: Potencia (W) desarrollada por el procesador durante cada compilación del AE en función del compilador y del nivel de optimización. (a) Clang (b) Visual C++

diferencias entre ambos compiladores para el mismo nivel de optimización –ya sean a favor de uno o de otro– son estadísticamente significativas con $\alpha = 0.05$). Más concretamente, Clang es más eficiente energéticamente con las opciones /Od y /O1, pero cae por debajo de Visual C++ con las opciones /O2 y /Ox. Esto ilustra el coste relativo de implementar cada nivel de optimización. De hecho, puede verse que en general y como es de esperar, tanto el tiempo de cómputo como la energía consumida aumentan al subir el nivel de optimización en ambos compiladores (aunque no hay diferencia estadísticamente significativa entre /O2 y /Ox). Sin embargo, la huella de procesamiento es muy diferente en cada caso: como muestra la Figura 2, mientras Clang mantiene un perfil esencialmente plano durante el proceso de construcción del AE ejecutable, la aplicación de los métodos de optimización de código en Visual C++ es mucho más pesada, elevando de manera dramática la potencia desarrollada (nótese que este esfuerzo intensivo no se refleja necesariamente en el rendimiento del ejecutable obtenido, tal como se ha visto anteriormente). Estas consideraciones son particularmente relevantes cuando se contemplan desde el prisma de la compilación como una actividad repetitiva dentro de las fases de desarrollo, depuración y rediseño del código, y cuyo impacto acumulado no puede obviarse.

Una última observación hace referencia al comportamiento térmico del procesador durante la ejecución del AE en los experimentos. Como se puede apreciar en la Figura 3, dicho comportamiento es claramente diferente entre las versiones obtenidas de cada compilador: mientras que para Visual C++ todas las versiones del programa mantienen la misma progresión térmica durante el tiempo que dura cada ejecución (y eventualmente se alcanzan las temperaturas más altas al final de la ejecución de la versión /Od, debido a su mayor extensión temporal), en el caso de Clang hay un calentamiento mucho más pronunciado en las versiones que emplean algún tipo de optimización, siendo el pico de temperatura en algún caso superior al alcanzado al final de la ejecución de la versión /Od. Esto parece contrastar con el perfil de potencia cualitativamente similar de ambos compiladores en relación con el empleo o no de optimización de código (Figura 1). Es interesante observar, no obstante,

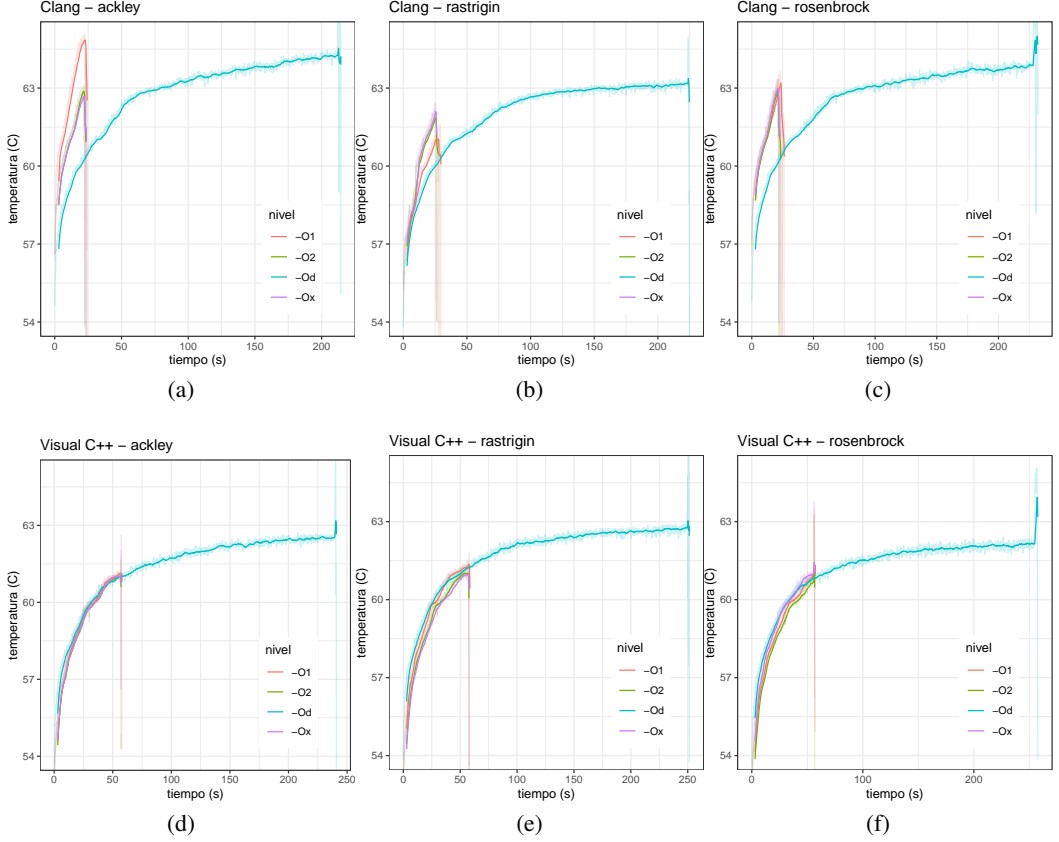

Figura 3: Temperatura del procesador durante la ejecución del algoritmo. La fila superior corresponde a Clang y la inferior a Visual C++. De izquierda a derecha en cada fila: ackley, rastrigin y rosenbrock.

que cuantitativamente la potencia desarrollada por el programa compilado con Visual C++ es algo inferior a la correspondiente para Clang (obsérvese la trayectoria de cada curva en relación con la cuadrícula de guía que mantiene la misma escala en todas las figuras). Esto apunta a la presencia de una no-linealidad en la disipación de calor en función del régimen de potencia en el que trabaja el procesador. Este tipo de fenómenos puede tener un impacto indirecto en la sostenibilidad del empleo del algoritmo, en la medida en la que un estrés térmico acumulado pueda conducir a una degradación más rápida de rendimiento a medio/largo plazo [12, 16].

## 4. Conclusiones

Aunque es habitual enfocar el análisis de la sostenibilidad y la eficiencia energética de los AEs desde una perspectiva puramente algorítmica, las decisiones tomadas durante la construcción del programa ejecutable pueden tener un impacto nada desdeñable en este contexto, tal y como se muestra en este trabajo. Aspectos tales como la elección del compilador o las opciones de optimización, a pesar de ser transparentes desde un punto de vista algorítmico (al menos de manera general – volveremos sobre este particular un poco más adelante), influyen de manera clara en la huella de procesamiento del mismo. En este sentido, la experimentación realizada con dos compiladores, Clang y Visual C++, muestra cómo uno de ellos (Clang) es notablemente superior al otro considerando las mismas opciones de optimización. Lógicamente, el funcionamiento interno de estas optimizaciones no resulta ser el mismo en ambos compiladores, aunque nominalmente sean análogas; esto es precisamente lo que el análisis determina y cuantifica. Es también reseñable que la desactivación de las optimizaciones produce un código ejecutable altamente ineficiente que, a pesar de no someter al procesador a una elevada carga de trabajo, resulta energéticamente desfavorable en su conjunto. Relacionado con esto, se ha podido identificar cómo el empleo de diferentes optimizaciones reduce el tiempo de cómputo a

costa de someter al procesador a un régimen de potencia más alto. Si bien el balance energético sigue siendo favorable en general en este caso, esto también plantea la posible existencia de externalidades asociadas: someter al procesador a un mayor estrés de temperatura de manera continuada en el tiempo puede conducir a estrangulamiento térmico y a degradación del hardware. Huelga decir que resultará del mayor interés extender y confirmar los hallazgos experimentales sobre un conjunto más amplio de compiladores, incluyendo, por ejemplo, la *GNU Compiler Collection* [8] que puede ser especialmente interesante por la gama de optimizaciones de código que ofrece y por lo extendido de su uso (al punto de constituir un estándar en sistemas operativos basados en UNIX y sus derivados). Precisamente en relación con esto último, resultará capital considerar otros sistemas operativos basados en UNIX.

Es importante tener en cuenta que todos los factores estudiados, así como sus efectos, han de ser entendidos de manera acumulada en el tiempo. La reducción de las ineficiencias energéticas no es solo una cuestión de principio, sino que hay además cada vez más evidencias de que cambios en el comportamiento del usuario final pueden conducir a un ahorro energético significativo [2]. En esta línea, sería del mayor interés escalar el estudio realizado, ampliando el banco de pruebas y el número de escenarios considerados con objeto de cuantificar de manera más precisa tanto los costes energéticos directos como las posibles externalidades. Existen, por ejemplo, opciones de optimización más agresivas que las consideradas y que incluso son desaconsejadas en entornos de producción debido a que pueden llegar a afectar al resultado de operaciones en punto flotante y dar lugar a otro tipo de inestabilidades. Estudiar la robustez de los AEs en esta situación tendría gran relevancia. El objetivo último es ser capaces de proporcionar algunas directrices para abordar la experimentación con algoritmos evolutivos de la manera más sostenible.

## Acknowledgments and Disclosure of Funding

Este trabajo se ha realizado con el apoyo del Ministerio de Ciencia e Innovación a través del proyecto Bio4Res (PID2021-125184NB-I00 – `http://bio4res.lcc.uma.es`), así como de la Universidad de Málaga, Campus de Excelencia Internacional Andalucía Tech.

Los autores declaran no tener conflictos de interés.

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
