# OpenReview forum: "Consumo Energético de Algoritmos Evolutivos en C++: Impacto del Compilador y el Nivel de Optimización"
_MAEB/2025/Congreso — MAEB 2025_

### Official Review · Reviewer_Jy2v · 2025-03-17
**Poca relación con los temas de interés del congreso**

**Rating:** 1
**Confidence:** 4

**Review:**

Este estudio analiza el consumo energético de los algoritmos evolutivos en función del compilador utilizado y los parámetros pasados al mismo. Se comparan los compiladores Clang y Visual C++ con diferentes niveles de optimización. Aunque se evalúa el rendimiento de varios algoritmos evolutivos, el trabajo podría haberse realizado perfectamente sobre otro conjunto de programas. La única relación con los temas de interés del congreso es que el conjunto de datos incluye algoritmos evolutivos, pero esto no es relevante para el estudio, que podría haberse titulado "Consumo Energético de **programas** en C++: Impacto del Compilador y el Nivel de Optimización".

---

### Official Review · Reviewer_HsJm · 2025-03-18
**Estudio de la influencia del compilador y sus banderas de optimización en el consumo de un algoritmo evolutivo**

**Rating:** 4
**Confidence:** 5

**Review:**

El trabajo presenta un estudio sobre el impacto que tiene en el tiempo de ejecución y consumo de un Algoritmo Evolutivo tanto el compilador utilizado como las opciones de compilación disponibles en el mismo. Se consideran varios problemas de optimización numérica de alta dimensionalidad y dos compiladores distintos: Clang y Microsoft Visual C++.

Se reportan mediciones de consumo, tiempo, potencia y temperatura, todas relacionadas con la CPU del sistema, tomadas con la herramienta de Intel Power Gadget. Los resultados reportados son la media de 30 ejecuciones, y se ha aplicado el test de Wilcoxon para obtener confianza estadística en los resultados.

El trabajo es sencillo pero interesante, ya que es necesario conocer el consumo de los AEs para ser capaz de reducirlo. Se trata de un estudio meramente experimental, y no se trata de profundizar en el análisis de los resultados para encontrar los motivos de la eficiencia o ineficiencia de los códigos. Sería interesante conocer información como el tamaño de los ejecutables, el uso de memoria y de otros recursos del HW.

También sería conveniente hacer un estudio sobre la incertidumbre del sistema, ya que se puede tratar de un entorno con una elevada incertidumbre que pudiera hacer que 30 ejecuciones independientes fueran insuficientes

---

### Official Review · Reviewer_3APU · 2025-03-19
**Interesante enfoque... aunque está en un estado un poco preliminar**

**Rating:** 4
**Confidence:** 4

**Review:**

En este artículo se analiza el impacto del compilador y las opciones de optimización en el consumo energético de algoritmos evolutivos en C++, utilizando dos compiladores determinados en la implementación de los algoritmos. Se trata de un trabajo interesante, que se encuadra dentro de la tendencia de analizar alternativas verdes a los algoritmos que utilizamos en la resolucion de problemas. El trabajo está bien redactado y estructurado, si bien hay algunas cuestiones que podrian ser mejoradas...

- En primer lugar, no veo justificada la eleccion del número de compliadores empleados, ni la eleccion concreta de dichos compiladores. Ademas, todo el trabajo se centra en entornos asociados al sistema operativo Windows, cuando la mayoría de las aplicaciones de cómputo se están realizando en entornos UNIX. Creo que habría que justificar la motivacion de este punto.
- Por otra parte, el número de pruebas realizadas me parece un poco escaso para alcanzar conclusiones suficientemente robustas. Supongo que la idea es extender el trabajo y más adelante publicar resultados más elaborados. Aún así me parece que podría haberse hecho algo más denso.

Por lo demás, me parece un trabajo interesante,  un buen comienzo para una investigación ulterior.

---

### Decision · Program_Chairs · 2025-03-20

Accept